# High Performance Size Exclusion Chromatography-Inductively Coupled Plasma-Mass Spectrometry to Study the Copper and Cadmium Complexation with Humic Acids

**DOI:** 10.3390/molecules24173201

**Published:** 2019-09-03

**Authors:** Marta Radaelli, Elisa Scalabrin, Giuseppa Toscano, Gabriele Capodaglio

**Affiliations:** Department of Environmental Sciences, Informatics and Statistics Ca’ Foscari University of Venice, Via Torino, 155, 30172 Mestre-Venezia, Italy

**Keywords:** humic acids, HPLC-ICP-MS, ASV, cadmium, copper, metal complexation

## Abstract

Dissolved organic matter (DOM) plays an important role in the environment by influencing the transport and distribution of organic and inorganic components through different processes: the retention, mobilization, and bio-availability of potentially toxic elements (PTEs). The aim of the present study is to examine the dimensional characterization of humic acids (HA) extracted from soil matrix, as well as to analyze the metal distribution among different ligand classes. The molecular size distribution of the HA extract from soil showed three dimensional classes: 52 KDa, 4.5 KDa, and 900 Da. HPSEC-ICP-MS measurements demonstrated that the dimensional classes, relative to first two fractions, bind the largest part of metals. The complexing capacity of HA was evaluated to assess the pollutants mobility in the environmental system. In particular, cadmium (Cd) and copper (Cu) complexation was investigated due to the great concern regarding their bio-availability and toxicity in natural waters. The complexing capacity of HA solution (20 mg/L) was measured by titration using a high-performance size exclusion chromatography (HP-SEC) coupled to an inductively coupled mass spectrometry (ICP-MS). Results obtained by this technique are compared with those obtained by anodic stripping voltammetry (ASV) to investigate the effects of kinetic lability of complexes on measurements carried by HPSEC-ICP-MS. In this study, results of ligand concentrations and stability constants obtained via the two techniques are assessed considering the detection window associated to the applied analytical methodology. Results obtained using the two analytical techniques showed that Cd is complexed by two classes of ligands. However, the ligand concentration values obtained using the two techniques are different, because the detection window associated to the two methodologies; the complexing capacity, which was obtained as sum of the two classes of ligands, were 33 nmol/L and 9 nmol/L for ASV and HPSEC-ICP-MS, respectively. The copper complexing capacities determined by the two methodologies are comparable: 166 and 139 nmol/L for ASV and HPSEC-ICP-MS, respectively. However, the results of Cu titration differ for the two techniques, highlighting only one class of ligands when ASV was used, and two classes when HPSEC-ICP-MS was employed. Differences on results obtained by the two techniques are explained considering the kinetic lability of complexes; the results show that, differently from previous studies, also Cu complexes can be kinetically labile, if one technique with high reaction time is used, as well some cadmium complexes are sufficient stable to be determined by HPSEC-ICP-MS.

## 1. Introduction

The development of an adequate monitoring program for toxic elements and the determination of tolerable effects require knowledge on the interaction of metals with abiotic and biotic factors, on their bio-availability and the mechanisms of metal penetration into living organisms; larger part of these processes are directly dependent from the chemical speciation of elements [1]. Therefore investigations of biogeochemical migration of elements and of the role of organic matter are crucial in order to define metal toxicity.

A significant component of the organic carbon in the environment is the dissolved organic matter (DOM), which mainly consists of humic substances (HS). HS are complex, multi-disperse mixtures of organic compounds derived from plant, animal, and microbial material decay [2]. The chemical structure of HS is closely related to their origin, with an apparent continuous distribution of their physico-chemical properties (molecular mass, functional groups, solubility, hydrophobicity, and charge). One of the main characteristics of HS is their ability to interact with metal ions, oxides, hydroxides, mineral, and organic compounds that influence metal speciation, bioavailability, toxicity, and mobility in the environment [3,4]. Particularly, it is known that carboxylic and phenolic groups attached to HS are able to operate as effective ligands for metals [5]. Therefore, the ability of HS complexation and their surface-active properties can influence the processes that control the sedimentation, bioavailability, and speciation of metals in the environment, even if they are present at trace concentrations. It has been demonstrated that HS can contribute to the reduction of metal concentrations (e.g., aluminum, Cr(VI)) [6,7] and element toxicity in water (e.g., arsenic, zinc, lead) [8,9].

Cadmium (Cd) is an environmentally important metal that can be toxic even at very low concentrations, because it is bio-accumulated in aquatic organisms, it binds plasmatic proteins and red blood cells. Moreover, Cd is chemically similar to zinc and can replace it in biological systems. Therefore, Cd use in industrial processes has been restricted by the European Union’s Restriction on Hazardous Substances directive [10]; further, Cd concentrations monitoring has become mandatory. In this context, i.e., considering the presence of HS, a better knowledge of the binding relations of this element is essential to understand its speciation and, consequently, its mobility in the environment. 

Copper (Cu) represents an essential micro-nutrient being involved in important biological processes. However, because it is used in agricultural and industrial processes, it can be present in the environment at high concentrations and this can have ecotoxic effects; there is evidence that its toxicity is significantly reduced when bonded in stable complexes, which prevent its bioavailability [11]. These results are also confirmed from a study that reported how Cu complexation by natural organic matter decreases its toxicity or its internalization for several non-feeding marine organisms, such as larval stages of invertebrates [12]. A study carried out in seawater emphasized how dissolved organic components are important ligands for Cu. Further, this study described a possible competition between copper and iron and other trace metals [13]. Moreover, investigations carried out on complexation by HS of Cd, Cr, Cu, Ni, and Pb showed that fulvic acids and HA in water are binding 95% and 83% of Cu, and 67% and 66% of Cd, respectively [14]. 

According to the definition of homologous compounds introduced by Buffle [15], a set of homologous compounds is defined as a group of molecules with similar environmental roles or behaviors, e.g., FA and HA. In this case, the speciation, involves the determination of elements associated to a different group of homologous compounds able to complex cations with different physico-chemical properties [16]. These two groups have different characteristics, HA having lower molecular mass and higher oxygen content than FA, which instead presents higher molecular mass, a more heterogeneous composition, and resistance to biological and chemical degradation [17]. The cation complexing capacity of HS, as suggested from [18], changes with molecular mass, origin, and charge density. Due to the HS heterogeneous structures, it is more appropriate to describe the metal binding characteristics as a distribution of the number of the ligand types with different binding strengths, rather than to measure a single value for bulk HS [19].

The analysis of DOM-metal compounds still remains a challenge, due to low concentrations of metals in the environment and the different oxidation forms in which they can be found; this implies the use of very sensitive and selective analytical methods, which permit to determine the speciation of elements in environmental samples [4]. In this context, an automated voltammetric in situ profiling system (VIP) was realized for the analysis of dissolved metal in estuaries and coastal sea-waters [20,21]. However, different analytical methods may determine metal-organic complexes of different stability and concentration in natural waters, both because a variety of natural ligands may exist and because different methods may have different detection windows [22]. The labile fraction detectable by different techniques may derive from the partial dissociation of complexes during the measurement. Indeed, each technique has a specific reaction time (time necessary to obtain the analytical signal), ranging from few µsec to hours, producing different results deriving from kinetic processes [23,24].

There is evidence that kinetic parameters are relevant in toxicity studies. Therefore, information about kinetic lability of complexes becomes particularly important. Experimental Cu lability taken together with speciation calculations has provided evidence that the bio-available Cu and its cellular toxicity are directly related to the labile metal concentration [25]. By considering these observations, measuring organic complexation using any single method at any single detection window may give only partial information of the true concentration of organically complexed element and its environmental effects. Size exclusion chromatography (SEC) has been widely used to estimate the molecular size of humic substances [4,26]. In more recent metal speciation studies, inductively coupled plasma-low resolution mass spectrometry (ICP-LRMS) has been employed to determine DOM-metal complexes in natural waters [27]. The use of this technique proved to be useful to obtain experimental evidences of the metal binding by DOM under multi-elemental conditions at environmental trace levels [28]. Indeed, HP-SEC chromatograms provide continuous molecular mass distribution of metals due to the heterogeneous composition of DOM, which consists of humic substances. Laborda et al. used a functional approach to determine which metals are associated with different groups of homologous compounds, in addition to characterizing metal-DOM complexes in compost leachates through the deconvolution analysis [29]. Furthermore, they studied the mobilization of metals from compost as a function of pH and the molecular mass of the DOM complexes using the same approach [30].

Nowadays, many attempts have been conducted in order to characterize the metal-organic matter interactions, which combine different techniques and models [31]. The aim of this study is the dimensional characterization of HA extracted from soil matrix and the analysis of metal distribution among the different dimensional classes and the HA complexing capacity, of Cd and Cu, in order to clarify the role of organic matter in the mobility and fate of these metals in the soil. The interaction of metal-HA is studied using two different techniques characterized from different detection window and that possess adequate sensitivity and selectivity: (1) size exclusion chromatography (SEC) equipped with an UV detector and hyphenated with the ICP-LRMS; and (2) anodic stripping voltammetry (ASV). Two independent techniques that can give complementary information on essential aspects of metal speciation in natural matrices to assess their bioavailability and toxicity. The first technique is more related to thermodynamic parameters while the second one is more affected from kinetic processes. Results are compared to emphasize the potential kinetic dissociation of complexes, which changes the distribution between metal species affecting their environmental behavior.

## 2. Materials and Methods

### 2.1. Samples and Reagents

The HA sample (HA soil) was obtained from an agricultural area located in the periferical area of Mestre. The sample was freeze-dried and stored at −20 °C until analysis. The preparation of the HA solution followed the standard procedure reported by Piccolo [32]. Briefly, 1 g of the sample was dissolved with deionized water and treated with NaOH 1 M to reach pH 12. Then, the solution was filtered with Polytetrafluoroethylene (PTFE) filters (pore size 0.45 µm), instead of cellulose mixed esters membranes, to avoid possible contamination from the filters themselves [33]. Further, a phosphate buffered-saline solution was added (10mM (NH_4_)_3_PO_4_, 137 mM NaCl, and 2.7 mM KCl) until a final concentration of 100 mg/L of HA soil was attained.

In order to avoid contamination, ultrapure water (18.2 MΩ cm, 0.01 TOC) was produced using a Purelab Ultra System (Elga LabWater system, Veolia Water VWS, High Wycombe, UK) and used for all steps, including the dilutions of samples and the preparation of standards, *tris*(hydroxymethyl)aminomethane (THAM) (ACS reagent, ≥99.8% Sigma-Aldrich-Merck KGaA, Darmstadt, Germany) and ammonium phosphate, (NH_4_)_3_PO_4_, were used as mobile phases.

### 2.2. HPSEC-UVvis-ICP-MS System

To analyze the size distribution of dissolved HA, a high-performance liquid chromatograph (Agilent 1100 series, Agilent Technologies, Waldbroon, Germany) equipped with a double beam spectrophotometer (Perkin Elmer Lambda 40) and a size exclusion macropourous silica-based TSK-GEL G4000PWXL column (Tosoh Bioscience, Griesheim, Germany) was used. The flow rate of the eluent was set at 0.5 mL/min and the injection volume was 100 µL. A preliminary investigation was carried out in order to define the optimal wavelengths for UV detection, the column to be used, and the composition of the mobile phase. The wavelength to detect the dissolved organic matter was λ = 270 nm, because it was the best compromise between the absorption of HA and the minimal signal contribution from the eluent (Figure 1).

The chromatogram reported the molecular mass distribution of HA extracted from soil and was analyzed by a deconvolution program to identify the mass size fractions (Peakfit V.4, Jandel, Scientific, San Rafael, CA, USA) (see Figure 2).

To detect the elements associated to the dimensional classes, the liquid chromatograph was coupled with an ICP-LRMS (Agilent 7500 ORS—Agilent Technologies Inc., Santa Clara, CA, USA) equipped with an Octopole Reaction System (ORS). The instrument was fitted with a concentric nebulizer, a Peltier-cooled quartz spray chamber, and a quartz torch with a quartz injector tube. Before each experiment, instrumental performance was optimized in terms of torch alignment, nebulizer gas flow rate, RF power, and lens voltages. The tuning was carried out using an aqueous multielement standard solution containing 1ng/mL of ^7^Li, ^89^Y, ^205^Tl plus, and ^140^Ce (Ultra Scientific, North Kingstown, RI, USA) to estimate and minimize levels of double charge ions and oxide elements (for details about the instrumental parameters see the Appendix A).

The macropourous silica-based column was chosen because of its application in functional speciation of metal-DOM [29,30] and because it permitted a large pH range, including alkaline conditions. A solution of THAM 10 mM and (NH_4_)_3_PO_4_ 2.5 mM, in comparison to other tested solutions (THAM and (NH_4_)_3_PO_4_ in different ratio or ammonium nitrate), buffered at pH 8.5 (pH close to natural seawater) was chosen as a mobile in order to prevent the precipitation of humic substances in columns. This eluent composition provided the best peak separation in comparison to other tested solutions, as ammonium nitrate [30,34], Moreover, the ionization efficiency of the ICP was maintained, thus permitting us to obtain low noise values during MS analysis. Therefore, a low limit of detection for many metals was found.

The column calibration for the HA molecular masses was carried out using the molecular mass standards of polystyrene sulfonates (PSS, Sigma Aldrich^®^-Merck KGaA, Darmstadt, Germany) of 210, 4300, 6800, 17,000, 32,000, and 77,000 Da. These were prepared in mobile phase solution [35] (see Appendix A).

### 2.3. ASV System

Measurements obtained by Differential Pulse Anodic Stripping Voltammetry (DPASV) were performed to study the interaction between HA and metals by determination of ligand concentration and the conditional stability constants following the procedure previously described [36]. A multipolarograph (VA computrace 797, Metrohm Herisau, Switzerland) equipped with an automatic system for standards dosage (Metrohm 800 Dosino) was employed. The working electrode was a preformed Thin Mercury Film Electrode (TMFE), plated on a Rotating Glassy Carbon Disk Electrode (RGCDE), a Ag/AgCl/KClsat was used as a reference electrode.

### 2.4. Cadmium and Copper Complexation

The effects of HA on Cd and Cu metal speciation using ASV and HPSEC-ICP-MS techniques have been considered due to their environmental importance.

The study of complexation of both metals was carried out on soil HA solution (20 mg/L) prepared in THAM and (NH4)_3_PO_4_ (pH = 8.5). The same buffer solution was used as a mobile phase for SEC in order to have the same equilibrium conditions present during the HPSEC-ICP-MS analysis and to compare the results obtained by the two techniques. The complexing capacity of humic acids was obtained using the titration of purified HA soil solution 20 mg/L, which was initially analyzed to determine the Cd and Cu concentrations already present, as well as to calculate the correct complexing capacity for the two metals. The total metal concentrations in the HA soil solution were determined using ICP-LRMS, with a previously described method [37]; the repeatability from 10 measurements was 3.7 and 0.65 nmol/L for Cu and Cd respectively and the limit of detection (LOD), calculated as 3 times the SD on blank measurements, was 0.52 and 0.097 nmol/L for Cu and Cd respectively.

The Cd and Cu titration was performed by increasing metal concentrations on the HA sample (from 0 to 300 µg/kg for both the metals), until the complete saturation of ligands. Solutions were analyzed using HPSEC-ICP-MS and ASV.

To estimate the complexed metal concentration obtained via HPSEC-ICP-MS, the metal/peak area ratio was calibrated by injection, in the chromatographic system, using EDTA (0.01 M) and 10 µg/kg of Cd or Cu. The precision of the measurements of complexing capacity was evaluated using three independent analysis of HA sample aliquots at the same metal concentration (35 and 1.3 nmol/L for Cu and Cd, respectively). The results, as standard deviation, were 3.25 and 0.26 nmol/L for Cu and Cd, respectively.

The complexing capacity of the HA solution for Cd and Cu expressed as ligand concentration and conditional stability constant, determined via DPASV, was performed following the previously described procedure [38,39,40,41]. In short, the ligands were titrated by the metal of interest. After each labile metal concentration [M’] was determined by the peak current, the organically complexed metal concentration was determined applying the mass balance (C_M_ − [M’]), where C_M_ is the total metal in the sample after each addition. The sample complexing capacity was estimated in terms of ligand concentration and conditional stability constant by plotting the ratio [M′]/(C_M_ − [M′]) versus [M′]. The linear shape of the plot of [M′]/(C_M_ − [M′]) versus [M’] indicates the presence of one single class of ligands complexing the element and one class of ligands characterized by a similar conditional stability constant. The concentration and the relative conditional stability constant can be calculated by the slope and intercept of the straight line obtained by a linear regression procedure [41]. When the plot took a curved shape, the metal was complexed by more than one class of ligands. Results when the model of two classes of ligands, i.e., the complexation parameters, were obtained by a nonlinear fitting procedure using the Marquart-Levenberg algorithm [36]; the elaboration was carried out via the SIGMA PLOT software, version 11.0 (^©^Systat Software, Gmbh, Erkrath, Germany).

Precision for the ASV method was tested as standard deviation of three replicate measurements of complexing capacity of the HA solution; results expressed as CV% were 10 and 1 for Cu and Cd, respectively. The accuracy of complexing capacity was measured by 3 titrations of one EDTA solution 1.00 μmol/L with Cu, the mean value was 0.96 μmol/L, SD 0.03 μmol/L.

The complexing capacity of the HA solution for Cd and Cu carried out using HPSEC-ICP-MS was obtained by the following procedure: after each addition of metal at the HA solution, the metal bound concentration ([ML]) was obtained by the peak area of the complexed metal and calibration value of metal/peak area ratio described above. After each metal addition, from the mass balance between the total metal concentration (C_M_) and the complexed metal concentration, the labile concentration [M’] was found; therefore, the ratio was calculated [M’]/[ML]. The complexing capacity and the relative conditional stability constant were calculated using the same procedure described above for the ASV titration.

## 3. Results and Discussion

### 3.1. HA Characterization Using SEC-UV-ICP-MS

The molecular size distribution of HA extracted from soil is shown in the HPSEC-UV chromatogram reported in Figure 2. The HA chromatogram showed three dimensional classes: 52 KDa (peak 1), 4.5 KDa (peak 2), and 900 Da (peak 3). The peak number 4 cannot be attributed to a HS because of its small dimension, as it is lower than the inferior limit of the column used. The most evident band is observed at 13.2 min (peak 2). These results agree with previous studies in the soil [42] and in sediments reporting the prevalence of relatively low-molecular weight HS [43,44].

The metal signal intensities detected by the ICP-MS on the same sample is shown in Figure 3a. It demonstrates that the dimensional classes relative to peaks 1 and 2, present in the UV chromatograms, bind the largest part of metals. This is evident for Cu, nickel (Ni), and cobalt (Co). With respect to zinc (Zn), it can bind via HS fraction corresponding to the peak 3 attributed to the class with the lowest molecular weight. Lead (Pb) presents a small concentration. However, we can observe that it is distributed between three fractions. This distribution between fractions is in general agreement with results obtained in previous studies, where a higher affinity of Ni, Co, and Cu for 1–10 kDa HA is reported, while Cd, Zn, and Pb are complexed by more fractions of HA (30–50 kDa and 100–300 kDa) [34,42].

The Cd concentration, present in the HA soil solution, was not detectable. However, its addition to the HA sample, at a concentration 1 µg/L, evidenced its affinity for fraction 2 (dimensional class 4.5 KDa) (see Figure 3b). The distribution is different from the results of Bolea et al. [34], but it agree with the previous study of Grzybowski [45]. Indeed this study, carried out on HS (mean molecular weight lower than 1000 Da) isolated from Baltic seawater, highlighted that Cd complex only the lowest molecular fraction of humic substances [45].

The results emphasize that all HA fractions extracted from the soil, especially the class 2, complex the major part of metals, demonstrating a low selectivity. This outcome could be due to the presence of different structures with similar molecular size but different affinities for single elements which, in a complex organic mixture, gave rise to a similar behaviours. The only fraction that seems more selective than other fractions is the number 3 (900 Da) which presents the highest affinity for Zn.

### 3.2. Cd and Cu Complexation: Comparison Between ASV and HPSEC-ICP-MS Results

The results of complexing capacity obtained using HPSEC-ICP-LRMS and ASV techniques are compared to better define HA characteristics. Our interest was focused on Cd and Cu, which represent widely distributed and toxic environmental contaminants, in order to assess the effect of HA binding activity that potentially impact their mobility and bioavailability.

Table 1 reports the values of ligands concentration, C_L_, and the corresponding conditional stability constants, logK, obtained using titration data of the same HA solution (20 mg/L) using the two techniques.

The elaboration of the titration curves data of Cd (Figure 4) obtained via ASV and HPSEC-ICP-MS measurements evidenced the presence of two classes of ligands. However, substantial differences were observed for the ligand concentrations; the complexing capacity, obtained as the sum of the two ligands classes, were 33 nmole/L and 9 nmole/L, usingASV and HPSEC-ICP-MS, respectively. Comparable conditional stability constants were observed (Table 1).

The result of ligands concentrations complexing Cd obtained using the ASV technique are 16.9 and 16.5 nmole/L. The corresponding conditional stability constant, reported as logarithmic values, are 8.31 and 6.97, respectively. The concentrations obtained via HPSEC-ICP-MS for the two ligands classes are 0.95 and 8.06 nmole/L, while log Ks are 8.02 and 6.18, respectively.

The elaboration of the titration curves data of Cu (Figure 5) obtained via ASV and HPSEC-ICP-MS shows a comparable complexing capacity for the HA soil solution (166 and 139 nmol/L, respectively). However, it must be underlined that ASV showed only one class of ligands while HPSEC-ICP-MS identified two classes (Table 1). The relative logarithmic values for the conditional stability constants were 9.52 and 6.60, respectively.

Results obtained using the two techniques must be assessed in consideration of the characteristic reaction times of the two methods. The permanence of compounds in the diffusion layer typical of ASV measurements takes some µsecs [23] while the residence time of complexes in the chromatographic column is more than 10 min. If the kinetic dissociation constant of complexes is high enough, a partial decomposition of complexes during the reaction time is observed. This produces a significant reduction of the detected conditional stability constant (K) [23]. Consequently, if the product Cl×K that define the detection window (where Cl is the ligand concentration and K is the conditional stability constant) is below the characteristic value for the used methodology, no metal complexation is observed [22]. Indeed, there are evidences that complexes of Cd with natural organic ligands in water are partially decomposed during ASV analysis [23]. During chromatographic separations, in which the reaction time is remarkably higher than in ASV, a significant part of complexes can be completely dissociated and the complexation by this class of ligands is not observed. Indeed, the complexing capacity measured via HPSEC-ICP-MS is significantly lower than ASV measurements; only complexes which have enough stability, from the kinetic point of view, are detected.

Analogous consideration must be made to analyze results obtained for Cu complexation. Measurements carried out using electrochemical techniques highlight that this element forms stable complexes not significantly affected from dissociation during the measurement [23]. However, studies of van den Berg and Donat clarify that kinetic lability of complexes must always be considered [22]. If the reaction time associated to the analytical technique is comparable to the kinetic dissociation constant of complexes, the effect is to reduce the value of the conditional stability constant obtained [23]. Based on these considerations, it is possible to state that ASV shows one single ligands class because complexes are characterized by similar conditional stability constants [41]. The measurements carried out using the chromatographic technique produce a partial dissociation of one class of ligands that reduce the corresponding conditional stability constant, the result leads to the identification of two ligand classes: the first is more stable while the second is partially labile. The complexation of Cu by two ligands classes is also confirmed via the chromatographical technique carried out on the HA solution (Figure 3a), which shows that the Cu is distributed between two organic fractions with a molecular weight of 52 and 4.5 KDa, respectively. This confirms that humic acids are present in different fractions characterized by different physical and chemical properties.

The complexation of Cu by more than one ligand was already reported in other studies [46,47,48]. For example, Muller determined the presence of three ligands in the sediments of the Arran Deep (Scotland) [49].

Our logK values, obtained via the ASV technique, are included in the range of values usually presents in the literature for HS of different typologies [42,45,48,49,50]. However, the stability constants reported in those studies are generally higher than the values obtained in our study, probably due to the minor abundance of aromatic groups in HA from agricultural soil. In this study, the presence of ligands that form kinetically stable Cd complexes was detected in comparison with results obtained in studies regarding marine environments. This result could be explained by the higher affinity of Cd for aliphatic and carboxylic groups, which probably constitutes the major part of soil HA employed for this study [51,52,53].

## 4. Conclusions

In this study, the dimensional characterization of HA extracted from soil was carried out. HA complexing capacity for metals and the distribution among different ligand classes were studied using two independent techniques: size exclusion chromatography (SEC) equipped with a UV detector and hyphenated with the ICP-LRMS; and the anodic stripping voltammetry (ASV). The results show that it is possible to perform a qualitative analysis of metal complexation and quantify the complexing capacity using the two techniques. Moreover, our study shows that the two techniques are also able to identify more than one classes of ligands. With respect to the data obtained for Cu and Cd speciation, both methods allowed the identification of two classes of ligands complexing these metals. However, results of ligand concentrations and stability constants obtained by the two techniques must be carefully assessed considering the detection window associated to the analytical methodology used. The ASV technique is characterized from a reaction times of a few µsec, in relation to the permanence of complexes in the diffusion layer. Therefore, it permits to determine also relatively labile complexes from the kinetic point of view. On the contrary, size exclusion chromatography, with a reaction time of a few minutes (time of permanence in the chromatographic column), permits the detection of only kinetically stable complexes.

Indeed, the size exclusion chromatography (SEC) equipped with an UV detector hyphenated with ICP-LRMS technique underestimated the concentration of ligands complexing Cd in comparison to the results obtained by the ASV methodology. The complexing capacity, obtained as the sum of the two classes of ligands, was 33 nmol/L and 9 nmol/L by ASV and HPSEC-ICP-MS, respectively. For Cu, which forms normally stable organic complexes, the results obtained using the two techniques were comparable. The total complexing capacity was 166 and 139 nmol/L for ASV and HPSEC-ICP-MS, respectively. The partial kinetic dissociation of complexes obtained using HPSEC-ICP-MS highlights the presence of more ligand classes, as indicated by the results of complexing capacity and by the chromatographic analysis. The results of the latter methodology showed that Cu was complexed by two classes of ligands with different molecular weight: 52 and 4.5 KDa, respectively.

Information about dimensional characteristics of humic matter is available when using HPSEC-ICP-MS and the complexing capacity obtained by the titration procedure. These methods permit a correlation of structural HA characteristics with their potential control on metal distribution and bio-availability. The HS complexation and bond strength are parameters that directly influence metal concentration and bio-availability, leading to different interactions with the organisms and the environment.

Results show that the ASV technique, when the evaluation of metal speciation in the environment is necessary, represents a more advisable methodology that obtains accurate information on the total complexing capacity. However, because there is evidence that kinetic parameters are also relevant in toxicity studies, information about kinetic lability of complexes become important too. Therefore, measurements of complexing capacity obtained using HPSEC-ICP-MS give essential information. Moreover, the SEC-ICP-MS technique permits the simultaneous determination of a high number of metals. Therefore, it is possible to carry out studies that assess the selectivity of ligands and competitive effects of complexation of different metals. Therefore, we can conclude that results obtained using the two independent methodology gave complementary information to assess the mechanisms involved in HA binding activity and metal mobility.

## Figures and Tables

**Figure 1 molecules-24-03201-f001:**
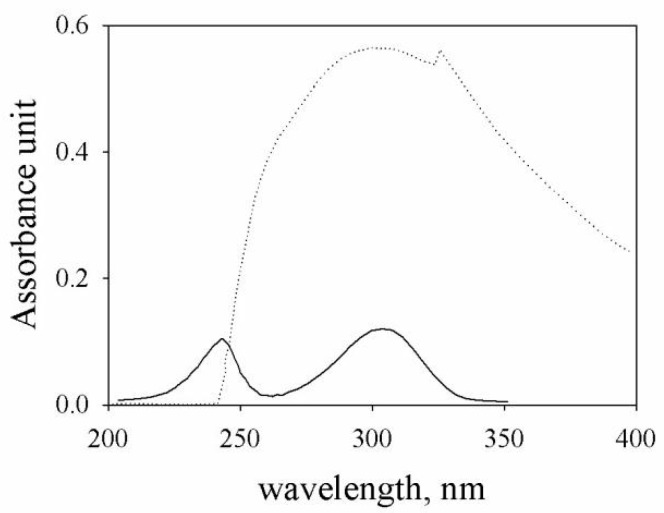
Ultraviolet (UV) absorbance spectrum of organic matter contained in humic acids (HA) soil (20 mg/L) (dotted line) and of eluent (continuous line).

**Figure 2 molecules-24-03201-f002:**
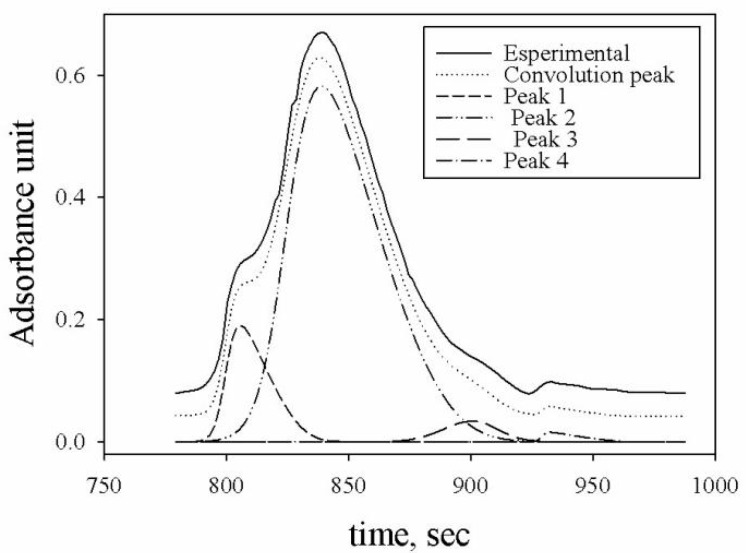
Size exclusion chromatography (SEC)-UV chromatogram: Molecular mass distribution of HA extracted from soil; peak numbers, obtained by the deconvolution program, represent the different dimensional classes of detected fractions, 52 KDa (peak 1), 4.5 KDa (peak 2), and 900 Da (peak 3), the peak 4 size cannot be attributed because of its dimension is lower than the inferior limit of the column (500 Da).

**Figure 3 molecules-24-03201-f003:**
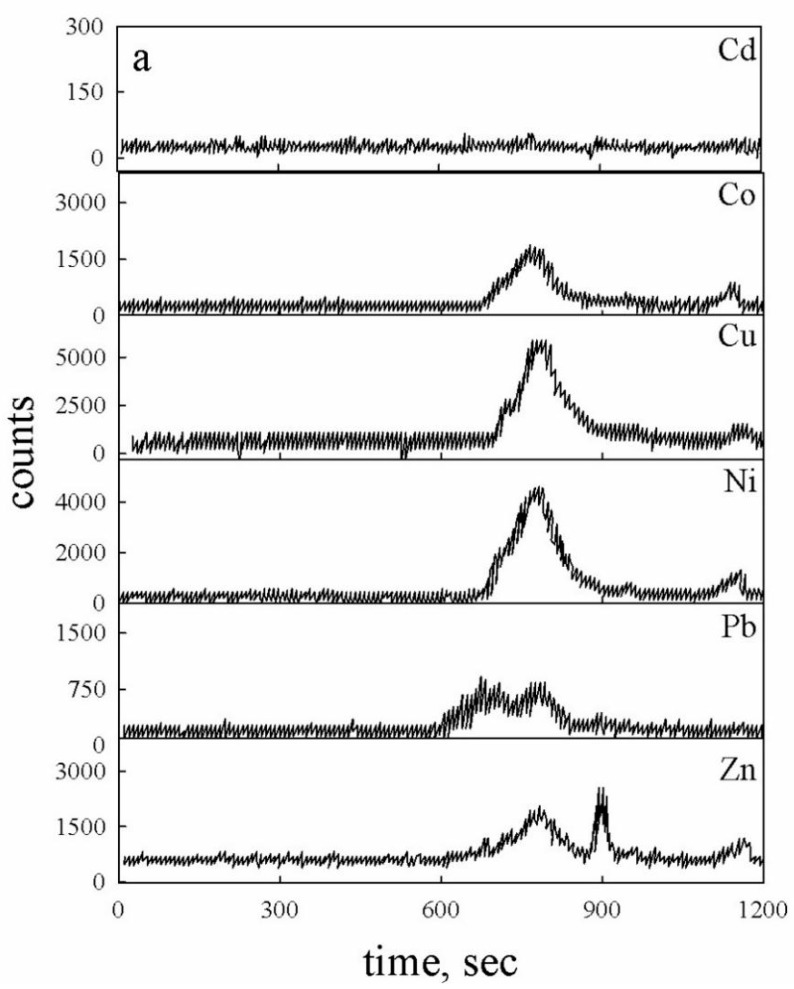
(**a**) High-performance size exclusion chromatography to inductively coupled mass spectrometry (HPSEC-ICP-MS) chromatogram for Cd, Co, Cu, Ni, Pb, and Zn. (**b**) HPSEC-ICP-MS chromatogram for Cd obtained after addition of 1 µg/L of standard solution to HA sample.

**Figure 4 molecules-24-03201-f004:**
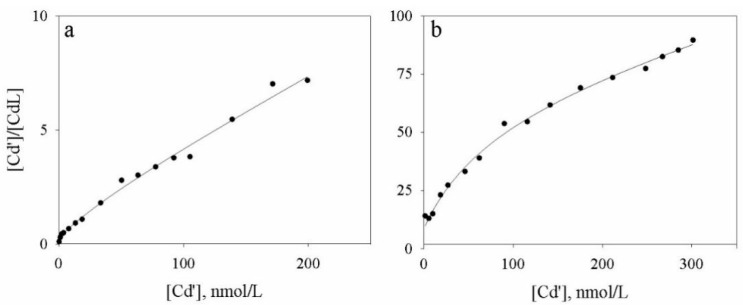
Plot of the elaborated data obtained from the ASV titration (**a**) and obtained from HPSEC-ICP-MS titration (**b**) for Cd.

**Figure 5 molecules-24-03201-f005:**
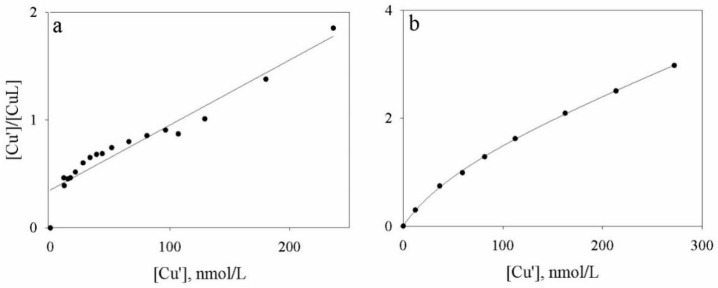
Plot of the elaborated data obtained from the ASV titration (**a**) and obtained from HPSEC-ICP-MS titration (**b**) for Cu.

**Table 1 molecules-24-03201-t001:** Concentration in nmol/L, Cl_s_, and corresponding LogK_s_ for the classes of ligands estimated for Cd and Cu, respectively, on the sample of HA 20 mg/L, extract from soil; values obtained via data elaboration obtained from titration followed by anodic stripping voltammetry (ASV) and HPSEC-ICP-MS, respectively.

Element		ASV	HPSEC-ICP-MS
Cd	Cl_1_	16.9	0.95
	logK_1_	8.31	8.02
	Cl_2_	16.5	8.06
	logK_2_	6.97	6.18
Cu	CL_1_	166	38.2
	logK_1_	7.23	9.52
	CL_2_		101.2
	logK_2_		6.60

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
