# Peer review of "High Performance Size Exclusion Chromatography-Inductively Coupled Plasma-Mass Spectrometry to Study the Copper and Cadmium Complexation with Humic Acids"

_molecules, 2019, doi:10.3390/molecules24173201_

Round 1

Reviewer 1 Report

The authors used HP-SEC-ICPMS and ASV to study the speciaiton of Cu and Cd that complexed with humic acids. The topic is novel, and the results are useful for analytical chemists to do Cu and Cd speciation analysis. The manuscript should be considered for publication after the authors addressed the following comments.

(1) The abstract needs revision to be more concise. I recommend authors to remove some background information and include more results.

(2) Introduction needs revision. The authors should highlight the imporance of studying metal-humic complexes and how their study will benefit to the environmental scientists. 

(3) The quality of all figures needs to be significantly improved. The current figures are not publishable.

(4) It is better to show some photos of analytical system in the method section, if it is possible.

(5) How did the authors control the data quality?

(6) There are many spelling and format errors in the manuscript, Please correct.  Such as Line 12, line 94, Line 132, line 263, 265, line 283, line 304, 307, 

Author Response

(1) The abstract needs revision to be more concise. I recommend authors to remove some background information and include more results.

The abstract was modified following the referee comment, background information are reduced and results are included

(2) Introduction needs revision. The authors should highlight the imporance of studying metal-humic complexes and how their study will benefit to the environmental scientists.

The introduction is significantly modified following the referee suggestions.

 (3) The quality of all figures needs to be significantly improved. The current figures are not publishable

The figure are improved, moreover the original figure in TIFF format are also included.

(4) It is better to show some photos of analytical system in the method section, if it is possible.

We do not think photos of the instrumentation can important important information, they are commercial instruments, anyway we added the scheme of the HPSEC-ICP-MS and one photo of the ASV in the supplementary material.

 (5) How did the authors control the data quality?

ICP-MS: The method precision on the measurements of complexed metal was evaluated by three independent analysis of HA sample aliquots at the same metal concentration; the CV% was used as precision parameter (as reported in the text lines 231-234)

About the total metal concentration determined by ICP-LRMS was applied a previously described method (Ranaldo et al. 2015) in which were reported repeatability and LOD

ASV: The complexing capacity measurements of the HA solution for Cd and Cu, expressed as ligand concentration and conditional stability constant, determined by DPASV, was performed following the previously described procedure and references are reported (Capodaglio et al. 1989, 1991, 1994; Scarponi et al. 1996).

Precision for ASV method was tested by three replicates measurements of HA, resulting in a CV% of 10 and 1 for Cu and Cd respectively. (as reported in the text lines 254).

Reviewer 2 Report

This an interesting paper on copper and cadmium complexation with humic acids. I think that the paper could be accepted after minor revision.

The paper should be revised by a native English speaker.

The pH was 8, then the hydrolysis of metals should be considered through the paper.

Author Response

As reported in the article the pH 8.5 as mobile phase was chosen because close to natural seawater, in order to prevent the precipitation of humic substances in column. Therefore the  hydrolysis of metals are the same present in natural water, do not require modelling to reproduce the natural system.

Reviewer 3 Report

The authors studied the interaction metal-HA by two different analytical techniques: HPSEC-UVvis-ICP-MS and ASV. This study seems to be interesting, however there is a lack of some information.

Introduction. The introduction is too extended, it is even longer that section results and discussion and I believe it should be shortened.

Experimental. I would prefer that authors extend the description of analytical methods, I miss information about HPLC parameters as well as ICP-MS working parameters.

Line 166. Could you please describe in detail Figure 2?

Abstract and lines 312-317. Authors explained that they applied there two analytical methods to compare results (Abstract) however it seems that obtained result can not be compared according to some information given by authors f.e. part of complexes can be completely dissociated during chromatographic separation. Why authors applied there two specific analytical methods? Are there any others that could be applied for this purpose?

Line 178. Was the application of mobile phase with this specific composition the original idea of authors or it was taken from literature?

Line 213 and 232. What was the metal concentration taken for precision calculation? 20mg/L?

The most important questions are: what are the advantages of using both techniques? What additional information did the authors get compared to those described in the literature?

Most lacking is the bulleting of novelties of this work. Please point it out.

Author Response

1_ Introduction. The introduction is too extended, it is even longer that section results and discussion and I believe it should be shortened.

The abstract was modified, background information are reduced and results are included, it was also included some comments on importance to study the kinetic lability of complexes.

Experimental. I would prefer that authors extend the description of analytical methods, I miss information about HPLC parameters as well as ICP-MS working parameters.

The analytical methods were previously published (references are present in the manuscript). Hovewer details of ICP-MS setting are submitted in the supplementary materials, Table S1.

Some details about the HPSEC are inserted (lines 163-164).

Line 166. Could you please describe in detail Figure 2?

As suggested the description of fig. 2 was done.

Abstract and lines 312-317. Authors explained that they applied there two analytical methods to compare results (Abstract) however it seems that obtained result can not be compared according to some information given by authors f.e. part of complexes can be completely dissociated during chromatographic separation. Why authors applied there two specific analytical methods? Are there any others that could be applied for this purpose?

As reported in the manuscript, difference on results obtained by different techniques are expected, comparison between results obtained by techniques with different detection window can give useful complementary information (see the introduction).

Line 178. Was the application of mobile phase with this specific composition the original idea of authors or it was taken from literature?

A solution of THAM 10 mM and (NH4)3PO4 2.5 mM provided the best peak separation in comparison to other tested solutions (THAM and (NH4)3PO4 in different ratio ), or ammonium nitrate. More over the use of (NH4)3PO4 was also suggested by the column specification (see lines 190-195)

Line 213 and 232. What was the metal concentration taken for precision calculation? 20mg/L

The concentration used were 35 and 1.3 nM for Cu and Cd respectively (see lines 233)

The most important questions are: what are the advantages of using both techniques? What additional information did the authors get compared to those described in the literature?

The information about kinetic lability of complexes is important for a correct interpretation of results in term of bioavailability. The modified introduction stress this concept, references are reported where importance of kinetic parameters are relevant in toxicology studies (see lines 105-116).

Most lacking is the bulleting of novelties of this work. Please point it out.

The aim of the work is described in more detail in the introduction, in particular lines 133-142. The novelties is related to use two independent techniques that can give complementary information of essential aspects in metal speciation in natural matrices to assess their bioavailabilty and toxicity. One technique more  related to thermodynamic parameters and the second more affected from kinetic aspects.